# The Analysis of Chlorogenic Acid and Caffeine Content and Its Correlation with Coffee Bean Color under Different Roasting Degree and Sources of Coffee (*Coffea arabica* Typica)

**Chia-Fang Tsai [1] and Irvan Prawira Julius Jioe [2,\*]**

[1] Department of Applied Cosmetology, National Tainan Junior College of Nursing, Tainan 70043, Taiwan; tsaicf@ntin.edu.tw
[2] Department of Biotechnology, TransWorld University, Douliu 64063, Taiwan
\* Correspondence: prawira_1705@yahoo.com; Tel.: +886-5-5370988

**Abstract:** Coffee is one of the main economic crops in the world and is now widely grown throughout Taiwan. The process of roasting coffee begins with the heating and smooth expansion of raw beans, which leads to changes in appearance and color while affecting the flavor and taste of coffee. So far, most coffee manufacturers have used visual inspection or colorimeter methods to identify differences in coffee quality. Moreover, there is no literature discussing the correlation of roasted bean color with caffeine and chlorogenic acid content. Therefore, the purpose of this experiment was to analyze the chlorogenic acid and caffeine content and their correlation with bean color under different roasting degrees and from different sources to establish basic data for the rapid identification of coffee quality in the future. In this experiment, the coffee *Coffea arabica* typica from Dongshan, Gukeng, and Sumatra's Indonesian rainforest was used, and the beans were roasted into four degrees: raw bean, light, medium, and dark roast, to investigate the appearance of the coffee beans and its correlation with caffeine and chlorogenic acid content. The results showed that with a higher roasting degree, caffeine content increased gradually, except for Indonesian beans, but the chlorogenic acid content in all samples showed a declining trend with the increase in roasting degree. The correlation between the chlorogenic acid content and the color space value of the coffee bean color shows that L\*, a\*, and h° in both ground and unground coffee are highly correlated. The C\* value of the ground and unground coffee showed a correlation coefficient of r = 0.159 ns and 0.299 ns, respectively. The correlation between the caffeine content and the color space value of the unground coffee bean shows that the a\*, b\*, and C\* value is highly correlated with the caffeine content. The color space values of ground coffee beans show no correlation with caffeine.

**Keywords:** *Coffea arabica*; caffeine; chlorogenic acid; coffee bean color

## 1. Introduction

Coffee is one of the main economic crops in the world, grown in more than 60 countries in tropical and subtropical areas [1], where the main varieties grown are arabica and robusta [2]. In 1884, coffee beans were introduced into Taiwan from the Philippines. These days, coffee has been widely planted in Taiwan in the Nantou Huisun Forest Farm, Gukeng in Yunlin, Zhongpu, Fanlu, Zhuqi, and Alishan in Chiayi, and Dongshan in Tainan [3]. According to the customs import and export data from 2018 to 2020, the import volume of coffee beans in 2018 was about 35,808 tons, about 37,719 tons in 2019, and about 41,507 tons in 2020, gradually increasing every year. Coffee beans contain high amounts of phenolic compounds [4] such as chlorogenic acid, caffeine, caffeic acid, etc., which can be used for dietary purposes [5]. Among them, phenolic compounds and chlorogenic acid have a high-content of about 6–12% in raw coffee beans [6]. Chlorogenic acid was discovered in coffee in 1837 [7] and has since been found to have antioxidant capacity [8–10], which can promote wound healing, reduce inflammation [11], and even prevent the occurrence of cancer

caused by DNA damage [12,13]. Many reports indicate that roasting temperature and grinding size can affect the amount of chlorogenic acid extracted from coffee beans [14,15]. Moreover, coffee that grows in different regions has different caffeine and chlorogenic acid contents [16]. Alica et al. [17] studied the caffeine content of coffee beans from 21 different regions and obtained the same result. Based on the research of Xu et al. [18], chlorogenic acids (CGAs) in coffee beans can be classified into three groups: caffeoylquinic acid (CQA), feruloylquinic acid (FQC), and dicaffeoylquinic acid (diCQA). Among these CGAs, caffeoylquinic acid, especially 5-caffeoylquinic acid (5-CQA), is dominant in coffee beans [19] and often used as a research subject due to its antioxidant capacity [11]

The process of roasting coffee begins with the heating and smooth expansion of raw beans, which leads to changes in appearances [20] and affects the flavor and taste of the coffee. The most used method in the coffee industry for roasting classification is the Agtron scale. Nevertheless, names such as cinnamon, regular brown, French espresso, and many others are frequently used in the coffee business [21]. Another index commonly used by coffee manufacturers is the colorimeter method CIE L*, a*, b*, C* value, and hue angle [22–24]. The CIE L* lightness value is used to identify the roasting degree of coffee; a low lightness value means a darker roasted coffee color [25]. Although some studies have discussed how different roasting levels affect the content of caffeine and chlorogenic acid [4,26], there is no information about the correlation between coffee color with caffeine and chlorogenic acid contents.

Therefore, the purpose of this experiment was to analyze the chlorogenic acid and caffeine content and their correlation with bean color under different roasting degrees and from different sources to establish basic data for the rapid identification of coffee quality in the future.

## 2. Materials and Methods

### 2.1. Coffee Beans Sources

This experiment used Coffea arabica typica from Dongshan (23°17′ N 120°26′ E), Gukeng (23°37′ N 120°35′ E), and Sumatra's Indonesian rain forest (0°59′ N 99°23′ E).

### 2.2. Roasting Degree

This experiment divided the three coffee sources into four roasting levels: raw bean, light, medium, and dark roast. The roasting temperature was set at 200 °C.

### 2.3. Coffee Bean Color Analysis

Unground raw bean and roasted coffee bean colors were analyzed with a colorimeter (HunterLab MiniScan EZ, 4000S)(Hunter Associates Laboratory, Inc., Reston, Virginia) to obtain CIE a*, b*, L*, C* values, and hue angles. The L* value refers to dark (0) vs. bright (100); the a* value refers to red (+) vs. green (−); the b* value refers to yellow (+) vs. blue (−); the C* value refers to saturation; the hue angle refers to the color. Each treatment had three replicates.

### 2.4. Sample Extraction

Ground coffee beans (0.2 g) were extracted with 20 mL of 70% methanol for 10 min. After filtration, the suspension was centrifuged at 13,000 rpm for 10 min and the supernatant was collected and stored in a refrigerator until required.

### 2.5. Chlorogenic Acid (5-CQA) Analysis

After the supernatant was filtered with a 0.45 μm filter membrane, the sample was analyzed with a high-performance liquid analyzer (HPLC) Nexera XR LC-20AD (Shimadzu Corporation, Kyoto, Japan), auto sampler Nexera XR SIL 21-ACXR (Shimadzu Corporation, Kyoto, Japan), and Diode Array Detector SPD-M30A (Shimadzu Corporation, Kyoto, Japan). For chromatographic separation, we used a Mightysil RP-18 GP 250-4.6 (5 μm) column. For the mobile phase, we used the following conditions: 1.18% phosphoric acid

(solution A) and acetonitrile (solution B) with gradient: 0–20 min: A (95%–87%) B (5–13%); 20–20.1 min: A (87–95%) B (13–5%); and 20.1–25 min: A (95–95%) B (5–5%) at a flow rate of 1 mL/min. Neochlorogenic acid was used as a reference standard and purchased from Sigma-Aldrich 94419.

### 2.6. Caffeine Analysis

The instrument analysis of caffeine content was similar to chlorogenic acid analysis. For the mobile phase, we used the following conditions: deionized water (solution A) and acetonitrile (solution B) with gradient: 0–10 min: A (85%–85%) B (15%–15%); 10–15 min: A (20%–20%) B (80%–80%); and 15–20 min: A (85%–85%) B (15%–15%) at a flow rate of 1 mL/min.

### 2.7. Statistical Analysis

Determination of roasting levels, bean colors, chlorogenic acid, and caffeine content in coffee extracts was performed in triplicate, and the mean values were calculated. The data were subjected to variance analysis and Duncan's test was used to assess the differences between means. A significant difference was presumed at a level of $p < 0.05$.

## 3. Results

### 3.1. Color Parameter Changes in Different Sources of Roasted Coffee Beans before and after Grinding

The results in Tables 1 and 2 reveal that the color parameters changed in different sources of green coffee beans. The L* value of both ground and unground green coffee showed a significantly higher content than roasted coffee. The color of Dongshan arabica raw coffee beans is close to the hex (8E7B58) color code; the color of Gukeng arabica raw coffee beans is close to the hex (846F4C) color code; the color of Indonesian arabica raw coffee beans is near the hex (918263) color code. However, after grinding, the color of raw coffee beans changes drastically, whereas Dongshan arabica raw ground coffee beans are close to hex (B09056) color code; Gukeng arabica raw ground coffee beans are close to ex (A88B5D) color code; Indonesian arabica raw ground coffee beans are near the ex (AE9678) color code. This difference can also be found in light- and medium-roasted coffee but both ground and unground dark roasted coffee showed little or no differences in appearance.

The comparison of coffee color before and after grinding in raw beans showed different color parameters, whereas the changes in the hue angle were more influenced by a* (with higher a* value in ground coffee). Furthermore, after roasting, the a* value of both ground and unground coffee increased drastically then slowly decreased as the roasting time was extended (Tables 1 and 2). Other parameters such as L*, b*, C*, and hue angle showed a downward tendency during the roasting process in both ground and unground coffee (Tables 1 and 2).

### 3.2. Chlorogenic Acid and Caffeine Content Changes before and after Roasting

The investigation of chlorogenic acid and caffeine contents of raw coffee beans from different sources showed that Indonesian raw coffee beans have the highest caffeine and chlorogenic acid contents of about 7.48 and 8.81 mg/g, respectively; while Gukeng and Dongshan raw coffee beans showed a slight difference in caffeine (5.06 and 4.35 mg/g, respectively) and chlorogenic acid (5 and 5.7 mg/g, respectively) contents (Figure 1A,B). However, after roasting, the caffeine content in both Gukeng and Dongshan coffee was slightly increased and reached the highest content in the dark roast condition (5.18 and 6.12 mg/g, respectively). Indonesian coffee showed the lowest caffeine content in light roasted (2.62 mg/g) and slightly increased in medium (4.1 mg/g) and dark (3.5 mg/g) roasting conditions. The chlorogenic acid content after roasting in all coffees showed a decline and reached the lowest content in dark-roasted coffee.

**Table 1.** CIE L*, a*, and b* values of different sources and roasting degrees of coffee beans.

| | Sources | L* | a* | b* | C* | Hue Angle | Common Names | Agtron/SCAA Classification | Nearest Hex Color | Real Color |
|---|---|---|---|---|---|---|---|---|---|---|
| **Seeds** | | | | | | | | | | |
| green coffee | Dongshan | 52.52 ± 2.66 a [z] | 2.35 ± 0.54 e | 21.88 ± 3.49 d | 21.97 ± 3.45 cd | 84.85 ± 1.97 b | green beans | | 8E7B58 | |
| | Gukeng | 47.99 ± 1.53 b | 3.34 ± 0.29 d | 22.57 ± 0.63 cd | 22.84 ± 0.67 c | 81.24 ± 0.47 c | green beans | | 846F4C | |
| | Indonesia | 55.04 ± 2.3 a | 0.84 ± 0.23 e | 19.16 ± 1.79 e | 19.18 ± 1.8 d | 87.5 ± 0.5 a | green beans | | 918263 | |
| light roast | Dongshan | 48.65 ± 0.54 b | 12.31 ± 1.06 a | 28.87 ± 0.59 a | 31.4 ± 0.87 a | 66.93 ± 1.53 e | light cinnamon | Tile #95 | 936943 | |
| | Gukeng | 46.42 ± 1.65 b | 7.14 ± 0.48 c | 24.97 ± 1.32 bc | 25.97 ± 1.37 b | 74.03 ± 0.59 d | light cinnamon | Tile #95 | 876944 | |
| | Indonesia | 42.19 ± 2.63 c | 13.27 ± 1.55 a | 18.4 ± 1.11 ab | 30.13 ± 0.79 a | 63.89 ± 2.58 f | cinnamon | Tile #85 | 825B46 | |
| medium roast | Dongshan | 36.59 ± 2.86 d | 9.71 ± 0.66 b | 18.15 ± 0.66 e | 20.8 ± 1.28 cd | 62.18 ± 0.45 f | regular brown | Tile #65 | 6E5039 | |
| | Gukeng | 36.05 ± 0.29 d | 7.61 ± 0.63 c | 18.15 ± 0.19 e | 20.34 ± 0.84 cd | 68.05 ± 0.96 e | medium | Tile #55 | 6A5038 | |
| | Indonesia | 34.9 ± 0.29 d | 9.78 ± 0.34 b | 17.02 ± 13.05 e | 20.62 ± 0.33 cd | 61.7 ± 0.55 f | medium | Tile #55 | 694C37 | |

**Table 1.** *Cont.*

| | Sources | L* | a* | b* | C* | Hue Angle | Common Names | Agtron/SCAA Classification | Nearest Hex Color | Real Color |
|---|---|---|---|---|---|---|---|---|---|---|
| | | | | **Seeds** | | | | | | |
| dark roast | Dongshan | 28.88 ± 0.26 e | 8.9 ± 0.14 b | 13.31 ± 0.82 f | 15.94 ± 0.67 e | 55.98 ± 1.73 g | espresso | Tile #35 | 583F30 | |
| | Gukeng | 21.5 ± 0.43 f | 7.42 ± 0.96 c | 9.53 ± 2.33 g | 10.25 ± 2.4 f | 51.66 ± 3 h | espresso | Tile #35 | 432F26 | |
| | Indonesia | 29.8 ± 0.55 e | 7.44 ± 0.4 c | 14.17 ± 0.27 f | 16.01 ± 0.32 e | 62.3 ± 1.28 f | espresso | Tile #35 | 584231 | |

Data are mean ± SD, *n* = 3. [z] Mean statistical data within column obtained by by Duncan's test at the 5% level. Same letters mean no significant.

**Table 2.** CIE L*, a*, and b* values of different sources and roasting degrees of ground coffee.

| | Sources | L* | a* | b* | C* | Hue Angle | Nearest Hex Color | Real Color |
|---|---|---|---|---|---|---|---|---|
| **Grounded** | | | | | | | | |
| unroasted coffee | Dongshan | 61.63 ± 2.39 ab [z] | 4.37 ± 0.13 g | 25.42 ± 0.5 cd | 25.79 ± 0.49 cde | 80.25 ± 0.34 ab | B09056 | |
| | Gukeng | 59.61 ± 0.17 b | 4.68 ± 0.79 g | 28.78 ± 4.24 bc | 29.1 ± 4.1 bcd | 81.44 ± 2.57 a | A88B5D | |
| | Indonesia | 63.35 ± 1.22 a | 4.56 ± 0.49 g | 19.13 ± 2.78 bc | 29.5 ± 2.7 bcd | 81.02 ± 1.54 ab | AE9678 | |
| light roast | Dongshan | 60.06 ± 1.22 b | 8.99 ± 0.3 e | 31.46 ± 0.16 bc | 32.72 ± 0.07 bc | 74.06 ± 0.59 c | B18A5A | |
| | Gukeng | 59.8 ± 0.26 b | 6.16 ± 0.18 f | 28.38 ± 0.17 bc | 29.04 ± 0.2 bcd | 77.75 ± 0.28 b | AB8B5E | |
| | Indonesia | 56.67 ± 2.74 c | 11.64 ± 0.12 d | 33.86 ± 0.72 b | 35.81 ± 0.64 b | 71.02 ± 0.54 c | AC7F4D | |
| medium roast | Dongshan | 42.65 ± 0.45 d | 14.95 ± 0.34 a | 32.61 ± 0.43 bc | 35.88 ± 0.59 b | 65.37 ± 0.24 d | 895A2F | |
| | Gukeng | 38.9 ± 1.08 e | 14.6 ± 0.23 a | 32 ± 0.58 bc | 35.330.59 b | 65.58 ± 0.28 d | 7E5227 | |
| | Indonesia | 33.67 ± 2.56 f | 13.3 ± 1.19 b | 41.02 ± 13.05 a | 30.01 ± 1.04 a | 70.94 ± 5.67 e | 704606 | |

**Table 2.** *Cont.*

|  | Sources | L* | a* | b* | C* | Hue Angle | Nearest Hex Color | Real Color |
|---|---|---|---|---|---|---|---|---|
|  | | | | **Grounded** | | | | |
| dark roast | Dongshan | 23.32 ± 0.71 g | 12.9 ± 0.03 bc | 19.98 ± 0.5 de | 23.94 ± 0.4 de | 57.36 ± 0.67 e | 51301A | |
|  | Gukeng | 16.1 ± 0.21 h | 11.7 ± 0.14 d | 14.94 ± 0.25 e | 19.28 ± 0.27 e | 51.88 ± 0.25 f | 3D2113 | |
|  | Indonesia | 17.68 ± 1.07 h | 12.2 ± 0.19 cd | 16.53 ± 0.51 e | 20.55 ± 0.52 e | 53.53 ± 0.44 f | 412414 | |

Data are mean ± SD, $n$ = 3. $^z$ Mean statistical data within the column were obtained by Duncan's test at the 5% level. Same letters mean no significant.

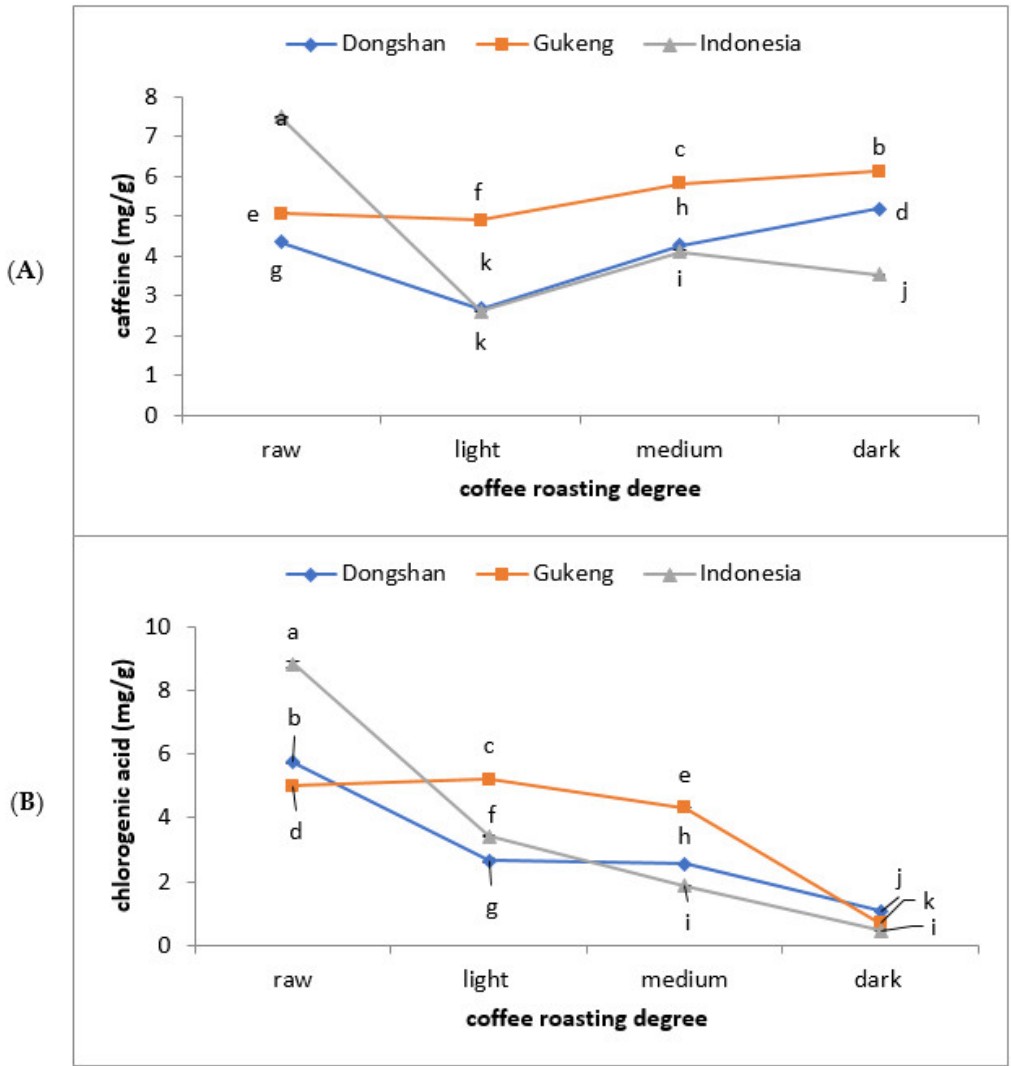

**Figure 1.** (**A**) Caffeine and (**B**) chlorogenic acid contents under different roasting degrees. Data are mean ± SD, *n* = 3. Mean statistical data within a column were obtained by Duncan's test at the 5% level (same letters means no significant).

*3.3. The Correlation between Chlorogenic Acid and Caffeine Content with CIE LAB Color Parameters*

The correlation between the chlorogenic acid and caffeine contents before and after grinding per the color parameters is shown in Figures 2–5. The chlorogenic acid content under different roasting degrees showed high correlation for parameters L* and a*, and hue angle both in ground and unground coffee (Figures 2 and 3). Correlation of the b* value in chlorogenic acid content in both ground and unground coffee did not show a similar tendency, whereas the b* value in ground coffee had a lower correlation coefficient® value (0.293 ns) than coffee seed (r = 0.44 **). The C* value in both ground and unground coffee also showed no correlation (r = 0.159 ns and 0.299 ns, respectively; Figures 2 and 3).

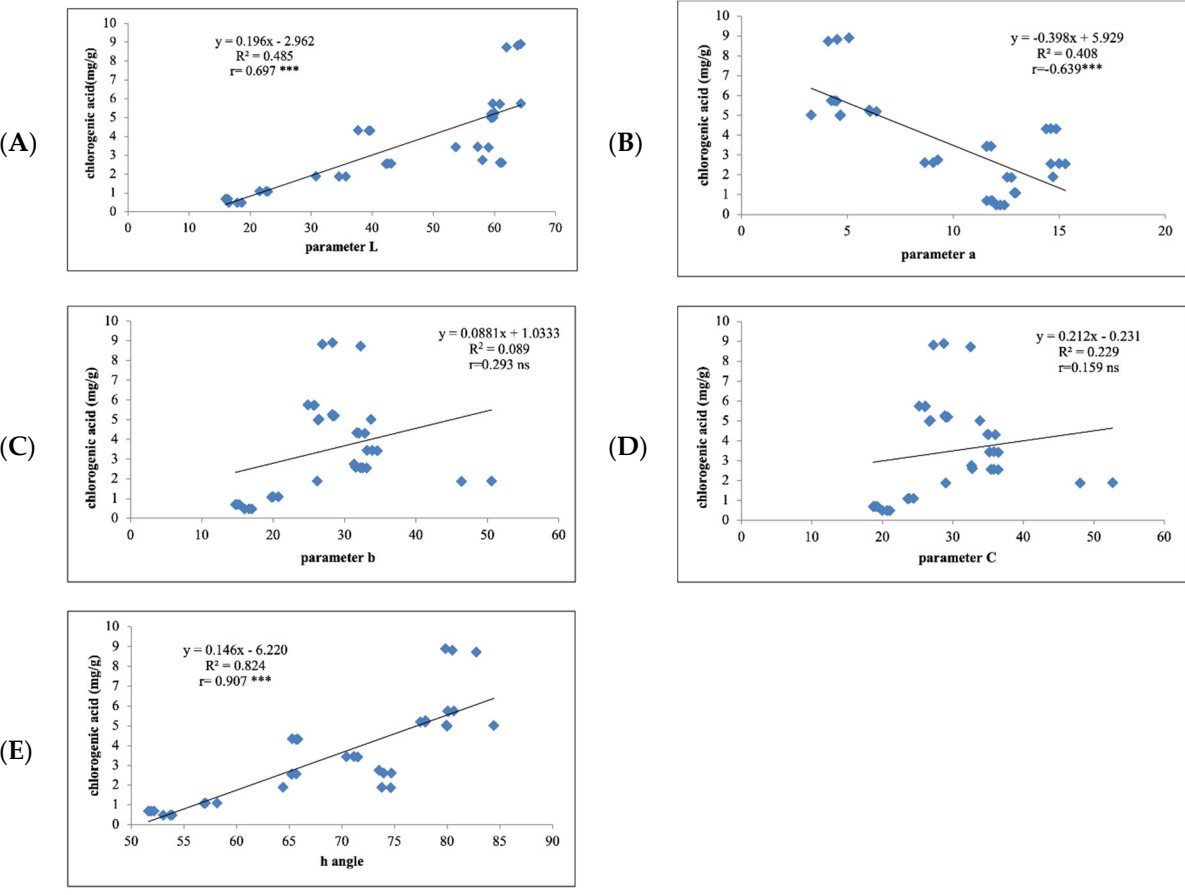

**Figure 2.** The correlation between ground roast coffee color (**A**) L*, (**B**) a*, (**C**) b*, and (**D**) C*, and (**E**) hue angle with chlorogenic acid content. (***: $p < 0.001$, ns: no significant).

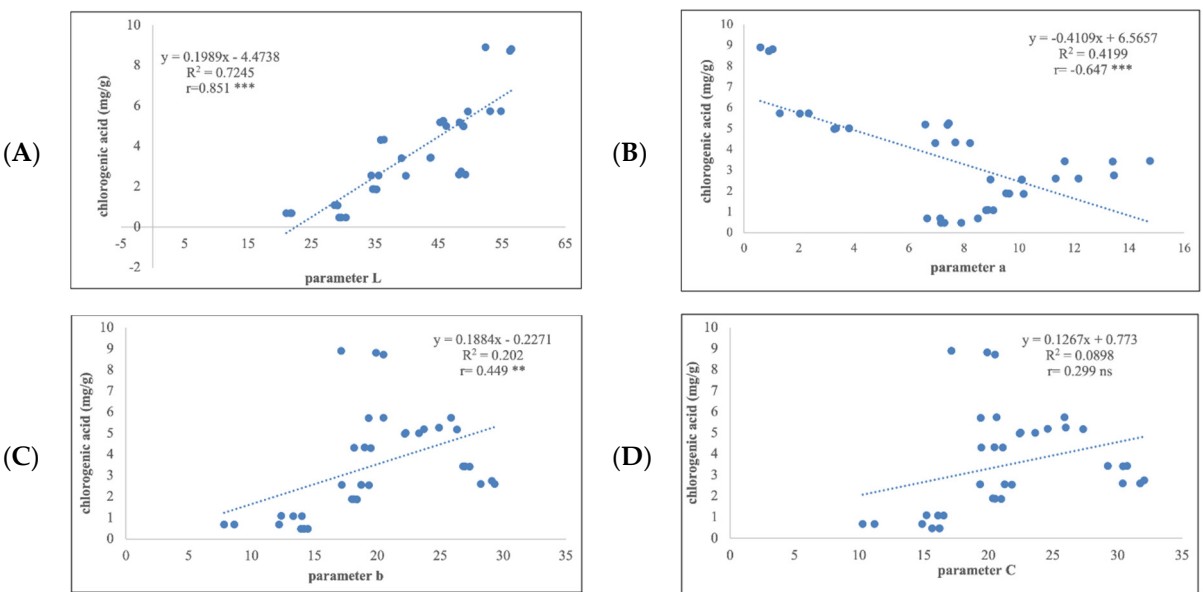

**Figure 3.** *Cont.*

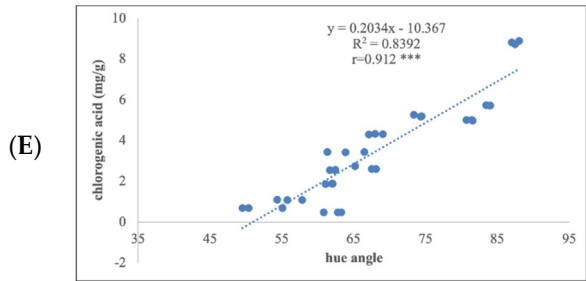

**Figure 3.** The correlation between roast coffee seed color (**A**) L*, (**B**) a*, (**C**) b*, and (**D**) C*, and (**E**) hue angle with chlorogenic acid content. (**: $p < 0.01$, ***: $p < 0.001$, ns: no significant).

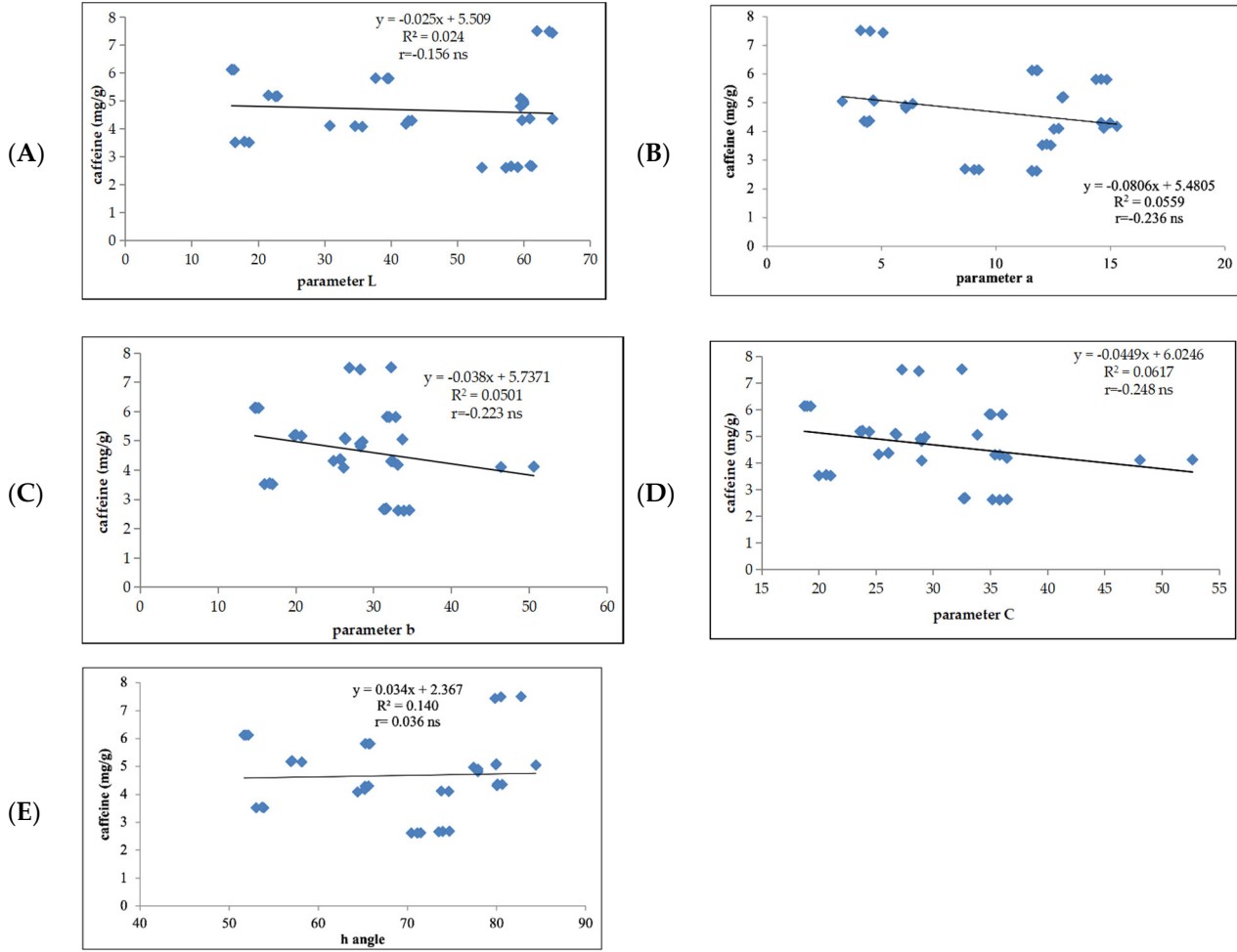

**Figure 4.** The correlation between ground roast coffee color (**A**) L*, (**B**) a*, (**C**) b*, and (**D**) C*, and (**E**) hue angle with caffeine content. (ns: no significant).

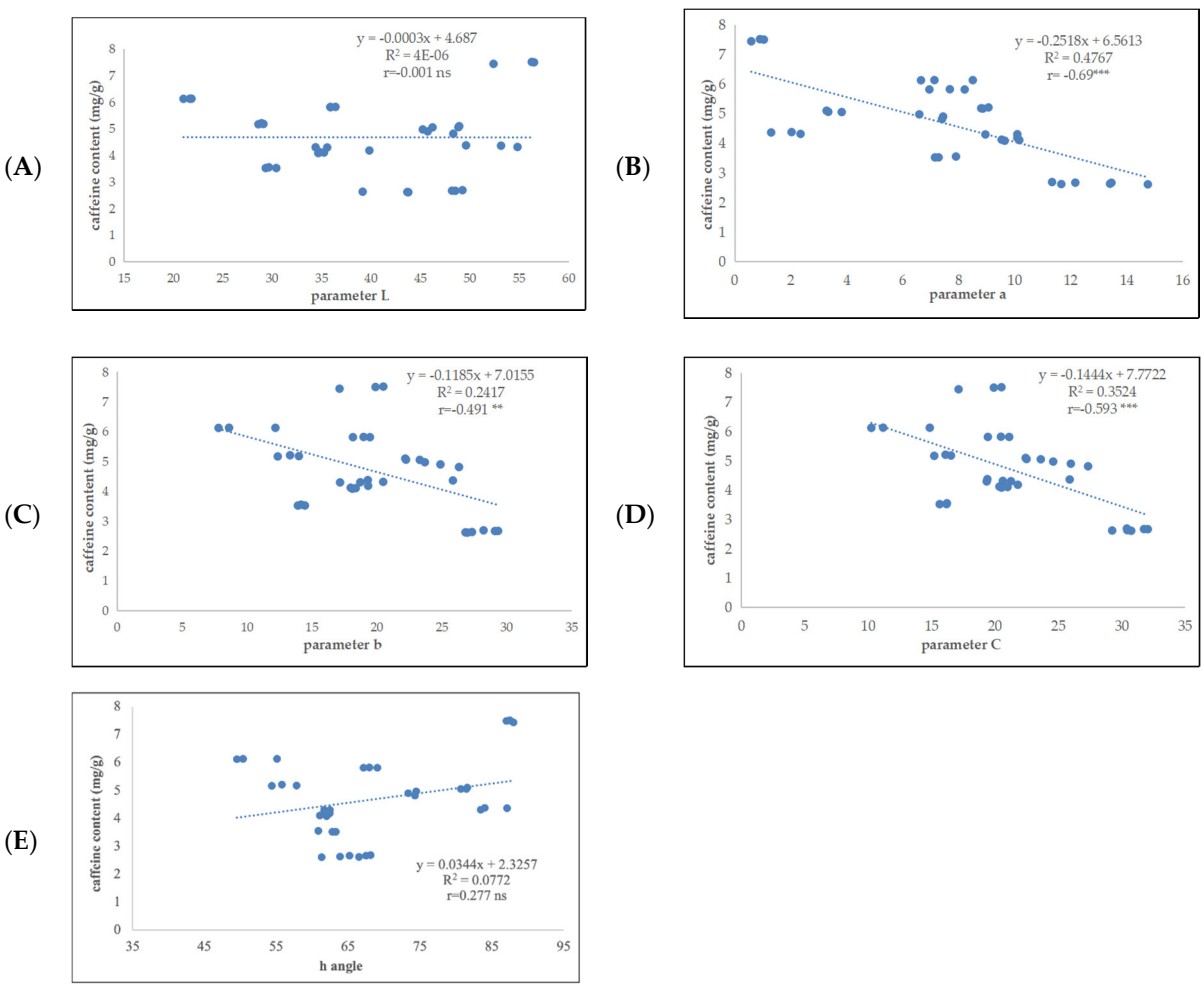

**Figure 5.** The correlation between roast coffee seed color (**A**) L*, (**B**) a*, (**C**) b*, and (**D**) C*, and (**E**) hue angle with caffeine content. (**: $p < 0.01$, ***: $p < 0.001$, ns: no significant).

The investigation of caffeine content using the CIE lab color parameter showed that ground coffee has no correlation with CIE lab color parameter (Figure 4). Meanwhile, the unground coffee's L* value and hue angle showed no correlation (r = −0.001 ns and 0.277 ns, respectively). The values of parameters a*, b*, and C* showed high correlation in unground coffee (r = −0.69 ***, −0.419 ***, and −0.593 ***, respectively; Figure 5).

## 4. Discussion

Colorimetrics are most commonly used to analyze the quality of fruits and vegetables [27–29]. Additionally, L*, a*, b*, C*, and hue angle (H°) parameters are mostly used in agriculture products [30,31], in which L* represents brightness; for a*, negative values refer to greenish and positive values refer to red; for b*, negative values refer to blue and positive values refer to yellow; C* stands for saturation; and H° stands for color attribute. The investigation into the color of coffee beans showed that the hue angle value decreased following the increase in the roasting degree (Tables 1 and 2) and showed similar results with previous literature [22,32,33]. In addition, during the research process, it was found that the raw coffee from different sources showed brownish colors (Table 1), a result that also appeared in the research of Bicho et al. [22], whereas raw bean arabica and robusta coffee showed brownish colors with the hex color codes (978873 and 99866D), Labch: 57.54, 2.02, 13.37, 13.53, and 81.47; and Labch: 56.93, 3.5, 16.19, 16.58, and 77.79.

According to the ISO 4149:2005 standard color of raw beans, there are five colors: bluish, greenish, whitish, yellowish, and brownish. The difference in the color of raw coffee

beans is related to the fermentation method used to remove seed mucilage [34]. There are two kinds of fermentation methods for removing mucilage from raw coffee beans: wet fermentation and dry fermentation. These two fermentation methods affect the color of the raw coffee beans. The dry fermentation method turns the raw coffee beans from white to brown, while the wet fermentation method turns them from blue to green [35,36]. Based on this research, a dry fermentation method might be used for the coffee beans in this experiment to remove the mucilage.

The a* value in green beans increased drastically then decreased slowly followed by the extension of roasting time, which showed a similar tendency to the trend reported in previous research [21,32]. Furthermore, comparison of the parameter a* value between ground and unground green coffee beans showed that unground beans had the lowest a* value, suggesting that the a* value in raw unground coffee beans might be related with a green color on its surface which degraded after roasting, resulting in the increase in a* value after roasting (Tables 1 and 2). L* and b* showed a downward tendency during the roasting process both in ground and unground coffee (Tables 1 and 2), which is concomitant with the research findings of Pittia et al. [37] and Budryn et al. [19]. In previous research, L* was mostly used to determine the roasting degree of coffee in which longer roasting time yielded a low L* value [25,38,39].

The correlation of chlorogenic acid and caffeine contents under different roasting degrees with CIE lab values was examined in this study.

Before roasting, the chlorogenic acid and caffeine contents in Indonesian coffee were significantly higher than in Gukeng and Dongshan coffee (Figure 1A,B). These differences might be related to the altitude of the cultivation area. In this research, Gukeng and Dongshan coffee were planted at 250–800 m, and Indonesian coffee, which showed high contents of caffeine and chlorogenic acid in raw coffee beans, was planted above 900–1400 m; these findings are similar to those in previous research [4,40].

The investigation of different sources of coffee beans under different roasting degrees showed that the caffeine slightly increased after roasting, showing a similar tendency to that reported in Hecimovic et al. [41], Moon et al. [4], and Ludwig et al. [42] (Figure 1A). However, the content of chlorogenic acid showed the opposite trend: with a higher roasting degree, the chlorogenic acid content decreased (Figure 1B), similar to Moon et al. [4] and Ludwig et al. [42]. Research indicates that with a longer roasting time, chlorogenic acid is converted into chlorogenic acid lactone [6]. Moreover, Dawidowiez and Typek [43] studied the accumulation of chlorogenic lactone at about 170–200 °C, and found that chlorogenic lactone content decreases at temperatures higher than 200 °C. The roasting temperature in this experiment was about 200 °C, and it is speculated that most of the chlorogenic acid in this experiment was converted into chlorogenic acid lactone.

In the coffee industry, in addition to using the SCAA/Agtron scale [21], the colorimeter CIE lab parameter is also used in coffee roast classification [22], especially the CIE L* lightness value, which was used to identify the roasting degree of coffee [25] and appeared comparable between this research (Tables 1 and 2) and previous research [32,44,45]. In addition, the CIE a* color value, which measures red and green color, showed a positive correlation with acrylamide content in roasted coffee [45]. The correlation between the chlorogenic acid and caffeine contents with color parameters was investigated in this research. The chlorogenic acid content decreased following the extension of the roasting degree (Figure 1B), which mirrors the findings of Moon et al. [4] and Ludwig et al. [42]. In addition, the high correlation between chlorogenic acid and parameter L*, a*, and hue angle in both ground and unground coffee (Figures 2 and 3) suggested that chlorogenic acid accumulation mostly occurs in the endosperm [46]. In contrast, a high correlation of caffeine content with b* and C* values was only found in unground coffee and showed no correlation in ground coffee (Figures 4 and 5). This suggested that caffeine accumulation occurs more in seed coats and cotyledon rather than in the endosperm [47] as the results of ground coffee beans showed no correlation with any color parameter (Figure 4). This

phenomenon can be found in other plants such as *Paullinia cupana* [48] and cocoa as well [49].

## 5. Conclusions

In this study, the assessment of chlorogenic acid and caffeine content in different sources of arabica coffee typica with variable roasting degrees showed the decline in chlorogenic acid content and increase in caffeine content after roasting. The CIE colorimeter method (L*, a*, b*, C* values) and hue angle were used as the common indices for roasting classification. The results indicated that chlorogenic acid content under different roasting degrees shows a high correlation with parameters L* and a*, and hue angle in both ground and unground coffee. Caffeine content in terms of parameters a*, b*, and C* showed a high correlation in unground coffee, while ground coffee showed no significant correlation with any parameter.

**Author Contributions:** C.-F.T. and I.P.J.J. conceived and designed the research. I.P.J.J. and C.-F.T. performed the experiments and analyzed all the data. I.P.J.J. and C.-F.T. wrote the paper. All authors have read and agreed to the published version of the manuscript.

**Funding:** This research received no external funding.

**Institutional Review Board Statement:** Not applicable.

**Informed Consent Statement:** Not applicable.

**Data Availability Statement:** The data presented in this study are available on request from the corresponding author. The data are not publicly available due to privacy.

**Conflicts of Interest:** The authors declare no conflict of interest.

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
