# Peer review of "The Analysis of Chlorogenic Acid and Caffeine Content and Its Correlation with Coffee Bean Color under Different Roasting Degree and Sources of Coffee (Coffea arabica Typica)"

_processes, doi:10.3390/pr9112040_

Round 1
Reviewer 1 Report
This paper investigated chlorogenic acid and caffeine content in different roasting process and evaluated its correlation with CIE lab parameters. The data presented are proper for this journal and do support the main conclusions driven. This reviewer just would appreciate to help improving coffee roasting processes by asking the authors to answer a few minor points raised below.
- Many studies have reported that there are several chlorogenic acids and derivatives in coffee beans and coffee roasting process is positively and negatively related with major chlorogenic acids and their derivatives. The authors described that chlorogenic acid increases during roasting process. It would be valuable that the authors describe chlorogenic acids and derivatives during roasting process and specifically denote a specific chlorogenic acid used in this study. Here, I recommend a recent reference summarizing overall chlorogenic acids and derivatives during coffee process.
Jiu Liang Xu, Tae Jin Kim, Jae-Kwang Kim, Yongsoo Choi, “Simultaneous roasting and extraction of green coffee beans by pressurized liquid extraction” Food Chemistry, 281 (2019) 261-268, Related page 264.
- In line 261~265, The message in this sentence is not clear and need to clarify this using different sentence expression or showing parameter number. For example, “ chlorogenic acid compare with CIE lab parameter showed the high correlation with roasting degree” where which parameter the author mentioned? Additionally, “whereas longer parameter L, a and hue angle,,,,” where “longer parameter” does not clear. Authors need to correct them, or rewrite the part.
Author Response
Thank you for your the comment. The detail response is in the attachment

Reviewer 2 Report
The presented topic is interesting and well prepared in terms of organization and selection of literature. The authors focused on aspects that fill the gap in knowledge about the chlorogenic acid and caffeine content and its correlation of beans color under different roasting degree.
The conducted study and analyzes are extensive and aimed at achieving the assumed goals.
In my opinion the manuscript meets the criteria for a scientific publication.
Author Response
Thank you for giving us good appraise in this manuscript.
We still made some correction manuscript due to other reviewers' comment. we will send the newest manuscripts as soon as we reply all the reviewers